# Prevalence of the metabolic syndrome in African populations: A systematic review and meta-analysis

**Arnol Bowo-Ngandji[1], Sebastien Kenmoe[2]\*, Jean Thierry Ebogo-Belobo[3], Raoul Kenfack-Momo[4], Guy Roussel Takuissu[5], Cyprien Kengne-Ndé[6], Donatien Serge Mbaga[1], Serges Tchatchouang[7], Josiane Kenfack-Zanguim[4], Robertine Lontuo Fogang[8], Elisabeth Zeuko'o Menkem[9], Juliette Laure Ndzie Ondigui[1], Ginette Irma Kame-Ngasse[3], Jeannette Nina Magoudjou-Pekam[4], Maxwell Wandji Nguedjo[5], Jean Paul Assam Assam[1], Damaris Enyegue Mandob[10], Judith Laure Ngondi[4]**

**1** Department of Microbiology, The University of Yaounde I, Yaounde, Cameroon, **2** Department of Microbiology and Parasitology, University of Buea, Buea, Cameroon, **3** Institute of Medical Research and Medicinal Plants Studies, Medical Research Centre, Yaounde, Cameroon, **4** Department of Biochemistry, The University of Yaounde I, Yaounde, Cameroon, **5** Centre for Food, Food Security and Nutrition Research, Institute of Medical Research and Medicinal Plants Studies, Yaounde, Cameroon, **6** Epidemiological Surveillance, Evaluation and Research Unit, National AIDS Control Committee, Douala, Cameroon, **7** Scientific Direction, Centre Pasteur of Cameroon, Yaounde, Cameroon, **8** Department of Animal Biology, University of Dschang, Dschang, Cameroon, **9** Department of Biomedical Sciences, University of Buea, Buea, Cameroon, **10** Higher Teachers' Training College, The University of Yaounde I, Yaounde, Cameroon

\* ken_sebas@yahoo.fr

## Abstract

### Background

The metabolic syndrome (MS) is a leading cause of death worldwide. Several studies have found MS to be prevalent in various African regions. However, no specific estimates of MS prevalence in African populations exist. The aim of this study was to estimate the overall prevalence of MS in the African populations.

### Methods

A systematic review was conducted in PubMed, Web of Science, Africa Index Medicus, and African Journal Online Scopus to find studies published up to the 15th of August 2022. Pooled prevalence was calculated based on six diagnostic methods. The pooled prevalence of MS was estimated using a random-effects model. Our risk of bias analysis was based on the Hoy et al. tool. A Heterogeneity ($I^2$) assessment was performed, as well as an Egger test for publication bias. PROSPERO number CRD42021275176 was assigned to this study.

### Results

In total, 297 studies corresponding to 345 prevalence data from 29 African countries and involving 156 464 participants were included. The overall prevalence of MS in Africa was 32.4% (95% CI: 30.2–34.7) with significant heterogeneity ($I^2$ = 98.9%; P<0.001). We obtained prevalence rates of 44.8% (95% CI: 24.8–65.7), 39.7% (95% CI: 31.7–48.1),

**Data Availability Statement:** All relevant data are within the paper and its Supporting Information files.

**Funding:** This project is part of the EDCTP2 programme supported by the European Union under grant agreement TMA2019PF-2705. The funders had no role in study design, data collection and analysis, decision to publish, or preparation of the manuscript.

**Competing interests:** The authors have declared that no competing interests exist.

33.1% (95% CI: 28.5–37.8), 31.6% (95% CI: 27.8–35.6) and 29.3% (95% CI: 25.7–33) using the WHO, revised NCEP-ATP III, JIS, NCEP/ATP III and IDF definition criteria, respectively. The prevalence of MS was significantly higher in adults >18 years with 33.1% (95%CI: 30.8–35.5) compared to children <18 years with 13.3% (95%CI: 7.3–20.6) (P<0.001). MS prevalence was significantly higher in females with 36.9% (95%CI: 33.2–40.7) compared to males with 26.7% (95%CI: 23.1–30.5) (P<0.001). The prevalence of MS was highest among Type 2 diabetes patients with 66.9% (95%CI: 60.3–73.1), followed by patients with coronary artery disease with 55.2% (95%CI: 50.8–59.6) and cardiovascular diseases with 48.3% (95%CI: 33.5–63.3) (P<0.001). With 33.6% (95% CI: 28.3–39.1), the southern African region was the most affected, followed by upper-middle income economies with 35% (95% CI: 29.5–40.6).

## Conclusion

This study, regardless of the definition used, reveals a high prevalence of MS in Africa, confirming the ongoing epidemiological transition in African countries. Early prevention and treatment strategies are urgently needed to reverse this trend.

## Introduction

Africa has always been known for its public health problems caused by infectious diseases and nutritional deficiencies; however, as a result of the nutritional transition and changes in lifestyle brought about by industrialization, chronic non-communicable diseases and over-nutrition have increased exponentially on the continent [1]. Aside from abdominal fat accumulation, hypertension, hypertriglyceridemia, hyperglycemia, insulin resistance, and low HDL-C levels are the leading causes of death worldwide [2]. The presence of three or more of these disorders in the same person gave rise to the concept of metabolic syndrome (MS) [3]. It increases the risk of type 2 diabetes by a factor of five, as well as the risk of cardiovascular disease by a factor of two to three [4,5]. It is critical to detect it early in populations because it contributes significantly to the occurrence of high-mortality diseases.

Diagnostic criteria for MS have been proposed by many groups, including the World Health Organization (WHO, 1998) [6], the European Group for the Study of Insulin Resistance (EGIR, 1999) [7], the American Association of Clinical Endocrinologists (AACE, 2003) [8], the National Cholesterol Education Program Adult Treatment Panel III (NCEP-ATP III, 2001) [9], the National Cholesterol Education Program Revised Adult Treatment Panel III (NCEP-ATP III Revised, 2005) [10], the International Diabetes Federation (IDF, 2005) [11], and a consensus between the AHA/NHLBI and IDF (Joint Interim Statement, 2009) [12]. There is agreement about the various components of MS, but there are differences in the details of the criteria used to describe them and their cut-off values [13].

None of these definitions have been established on a sample of African, but mainly Western, individuals, making it difficult to choose one of the definitions for studies done in Africa [14]. This sometimes justifies the use of several definitions. Previously thought to be uncommon in Africa [15], numerous studies have revealed higher prevalence in various Sub-Saharan African countries, most notably in Nigeria (23%) [16] and Ghana (41.8%) [17]; and in North African countries with prevalence of 39.6% and 48.5% observed in Tunisia and Morocco respectively [18,19]. Because of the rapid rise in MS, non-communicable diseases (NCDs) such

as T2DM, hypertension, and obesity are on the rise [20]. Indeed, it is predicted that by 2030, the burden of NCDs mortality will exceed that of communicable, maternal, neonatal, and nutritional diseases [21,22].

Overall, most regional systematic reviews on the prevalence of MS have been published in various regions of the world, including the Middle East [23], Asia-Pacific [24], low- and middle-income countries [25] and Latin America [26]. Several systematic reviews and meta-analyses on the prevalence of MS in Africa have been conducted in selected countries, with wide disparities and inconsistent results [27,28]. Many systematic reviews on the prevalence of MS have been conducted, primarily in Sub-Saharan Africa, and have focused on specific populations such as apparently healthy patients [14], patients with type 2 diabetes [29], people living with HIV [30] or people with mental illness [31] and have not included all available data on the continent's epidemiological situation of MS. As a result, there is a lack of systematic evidence on the prevalence of MS in many African countries, and there is no realistic picture of the continent's prevalence of MS. As a result, evidence from various populations and diagnostic definitions must be compiled.

To estimate the prevalence of MS in Africa, we conducted a systematic review and meta-analysis, focusing on diagnostic criteria, population groups, and regions. Similarly, the goal was to provide data on the estimated disease burden and trends so that targeted strategies for MS prevention and management in Africa could be planned and implemented.

## Materials and methods

### Registration

The Preferred Reporting Items for Systematic Reviews and Meta-Analyses (PRISMA) guidelines were followed for this systematic review (S1 Table) [32]. This review's protocol was entered into the International Prospective Register of Systematic Reviews (PROSPERO, n˚. CRD42021275176).

### Eligibility criteria

All available observational studies (cross-sectional, case-control and cohort (baseline) studies) that reported the prevalence of MS in the African population were eligible. There were no limitations on the target groups in terms of age and sex. These studies were carried out until August 15, 2022. The following studies were considered: (1) original studies, (2) human studies, and (3) studies published in French or English. Individual studies considered the presence of MS if it was defined by one of the following generally accepted criteria: World Health Organization (WHO, 1998), National Cholesterol Education Program Adult Treatment Panel III (NCEP-ATP III, 2001 and Revised NCEP-ATP III, 2005), American Association of Clinical Endocrinologists (AACE, 2003), International Diabetes Federation (IDF, 2005), and Joint Interim Statement (JIS, 2009). (Table 1). All studies in which the prevalence of MS was reported in a non-African country were excluded. Non-original research (such as reviews, editorials, case reports, case series, and letters or commentaries), unknown/unclear methods of diagnosing MS, studies without an abstract or full text, duplicates, and studies with fewer than ten participants were also excluded.

### Data sources and search strategy

A systematic search of electronic bibliographic databases for publications was conducted from inception to the 15th of August 2022. For advanced search strategies, PubMed, Web of Science, Africa Index Medicus, and African Journal Online Scopus were used (S2 Table). A thorough

**Table 1. Criteria for clinical diagnosis of the metabolic syndrome according to various definitions.**

| Criteria for diagnosis of MS | | WHO (1998) [33] | AACE (2003) [8] | NCEP-ATPIII (2001) [9] | Revised NCEP −ATP III (2005) [10] | IDF (2005) [34] | JIS (2009) [12] |
|---|---|---|---|---|---|---|---|
| | | insulin resistance together with two or more of the following: Diabetes diagnosis or FBG≥110 mg/dL or IR with ≥2 of the following | WC ≥102 cm (M) or ≥88 cm (F) with the presence of ≥2 of the following | Presence of any 3 of 5 of the following | Presence of any 3 of 5 of the following | WC:>94 cm (M);>80 cm (F) with the presence of ≥2 of the following | Presence of any 3 of 5 of the following |
| Hyperglycemia | Fasting glucose | Already required | ≥110 mg/dl or drug treatment for elevated blood glucose | ≥110 mg/dl | ≥100 mg/dL or on Rx for elevated glucose | ≥100 mg/dl or diagnosed diabetes | ≥100 mg/dl or diagnosed diabetes |
| Dyslipidemia | TG: | ≥150 mg/d | ≥150 mg/dl or drug treatment for elevated triglycerides | ≥150 mg/ dl | ≥150 mg/dL or on TG Rx | ≥150 mg/dl or on TG Rx | ≥150 mg/dl or on TG Rx |
| | HDL-C: | M: < 35 mg/dL F: < 39 mg/dl | ≤40 mg/dl (M) ≤50 mg/dl (F) or drug treatment for low HDL cholesterol | M: < 40 mg/dL F: < 50 mg/dL | M: < 40 mg/dL F: < 50 mg/Dl or on HDL-C Rx | M: < 40 mg/dL F: < 50 mg/Dl or on HDL-C Rx | M: < 40 mg/dL F: < 50 mg/Dl or on HDL-C Rx |
| Hypertension | Blood pressure | ≥140/90 mmHg | ≥130/85 mmHg or treatment for hypertension | ≥130/85 mmHg | SBP:≥130 mmHg or DBP:≥85 mmHg or on hypertension Rx | SBP:≥130 mmHg or DBP:≥85 mmHg or on hypertension Rx | SBP:≥130 mmHg or DBP:≥85 mmHg or on hypertension Rx |
| Obesity | WC | | | M:>102 cm F:>88 cm | M:≥102 cm F:≥88 cm | Already required | Ethnic dependent |
| | Waist/ hip ratio: | M:>0.9 F:>0.85 or BMI>30 kg/m$^2$ | | | | | |
| Other | | UAE≥20 µg/min | | | | | |

AACE: American Association of Clinical Endocrinologists, BMI: Body mass index; DBP: Diastolic blood pressure; F: Female; FBG: Fasting blood glucose; HDL-C: High density lipoprotein cholesterol; IDF: International Diabetes Federation; IR: Insulin resistance; JIS: Joint Interim Statement; M: Male; NCEP: National Cholesterol Education Program; Rx: Treatment; SBP: Systolic blood pressure; TG: Triglyceride; UAE: Urinary albumin excretion; WHO: World Health Organization; WC: Waist circumference.

review of the reference lists of all relevant articles was conducted to complete the search for potential additional data sources in the bibliographic databases.

## Study selection

Following the identification and removal of duplicates between the various bibliographic databases, the remaining titles and abstracts of the articles were independently reviewed by three authors (Kenmoe S, Ebogo-Belobo JT, and Bowo-Ngandji A) to determine the potential eligibility of the articles and finalize their full assessment. Data from the included studies were extracted by 18 of the study authors using a Google form and verified by Kenmoe S and Bowo-Ngandji A. For each included study, the name of the first author, the year of publication, the study design, the country, the country income level, the sampling method, the time of data collection, the study period, the age range of study participants, the percentage of males, the recruitment setting, the sample size, the study population, the diagnostic criteria for MS used, and the number of positive individuals for each diagnostic criterion were extracted. Any disagreements during study selection and data extraction were resolved through discussion and consensus, but if this did not work, the opinion of another author was required for final judgment.

## Quality assessment

To assess the quality of the studies, the methodological quality assessment tool for included studies developed for cross-sectional studies was used (S3 Table) [35].

## Statistical analysis

The prevalence of MS reported in the chosen studies was examined using the six diagnostic criteria. In the event of duplicate data, the study with the largest sample size was considered. In our review, only the diagnostic criterion with the highest sensitivity was considered in the overall estimate of the burden of MS for studies where the prevalence of MS was reported using more than two diagnostic criteria for the same population. The current meta-analysis was carried out with the statistical software R's "meta" package (version 3.6.2.) [36,37]. The prevalence of the pooled data was performed using a random-effects meta-analysis with a double arcsine Freeman-Tukey transformation [38,39]. The parameters $I^2$ (>50%), H (>1) and P-value of the Q-test (<0.05) were used to indicate significant heterogeneity [40]. By comparing the prevalence of MS in subgroups defined by several study characteristics, potential sources of heterogeneity were investigated. When pooling prevalence in the final data analysis, subgroups with fewer than three individual studies were excluded. The results were organized and classified using a variety of criteria, the most important of which were diagnostic criteria, population groups and regions in Africa (North Africa, West Africa, Southern Africa, East Africa, and Central Africa). Egger's test (P-value < 0.1) and funnel plot skewness were used to detect publication bias, and sensitivity analyses were performed on cross-sectional and low-risk-of-bias studies [41].

# Results

## Study selection

After searching the databases, a total of 9656 articles were found. Following the removal of the 1811 duplicates, the remaining 7845 articles were examined based on their titles and abstracts, with 7219 articles being excluded due to irrelevant titles and abstracts. Using the inclusion criteria, the full texts of 614 articles were downloaded and screened for inclusion. 317 articles were eliminated after reading for a variety of reasons. Finally, we examined 297 articles that met the inclusion criteria and contained data on the prevalence of MS in African countries in our meta-analysis (Fig 1) [42–338].

## Study characteristics

The articles chosen were published between 2004 and 2022, and the participants in the study were recruited between 1985 and 2021. (S4 and S5 Tables). Most participants were apparently healthy individuals (48.4%; 167/345), T2DM patients (13.3%; 46/345), and HIV-infected patients (9.3%; 32/345). This review included studies from 29 African countries, with the greatest number of studies coming from the following: Nigeria (n = 66), South Africa (n = 49), Ethiopia (n = 36), Ghana (n = 34), Egypt (n = 31), Tunisia (n = 29), Morocco (n = 18), and Cameroon (n = 13) are the countries represented. West Africa had the highest prevalence rate (32.5%; 112/345), followed by North Africa (27.3%; 94/345) and East Africa (17.7%; 61/345).

The rest came from Southern Africa (15.9%; 55/345) and Central Africa (6.7%; 23/345). Cross-sectional studies accounted for 318/345 (92.1%), hospital-based studies accounted for 248 (71.9%), prospective data collection and analysis accounted for 337 (97.7%), and non-probability sampling methods accounted for 284 (82.3%) of the prevalence data. In terms of MS diagnostic criteria, 115 studies (33.3%) used the IDF 2005, 109 (31.6%) used the NCEP

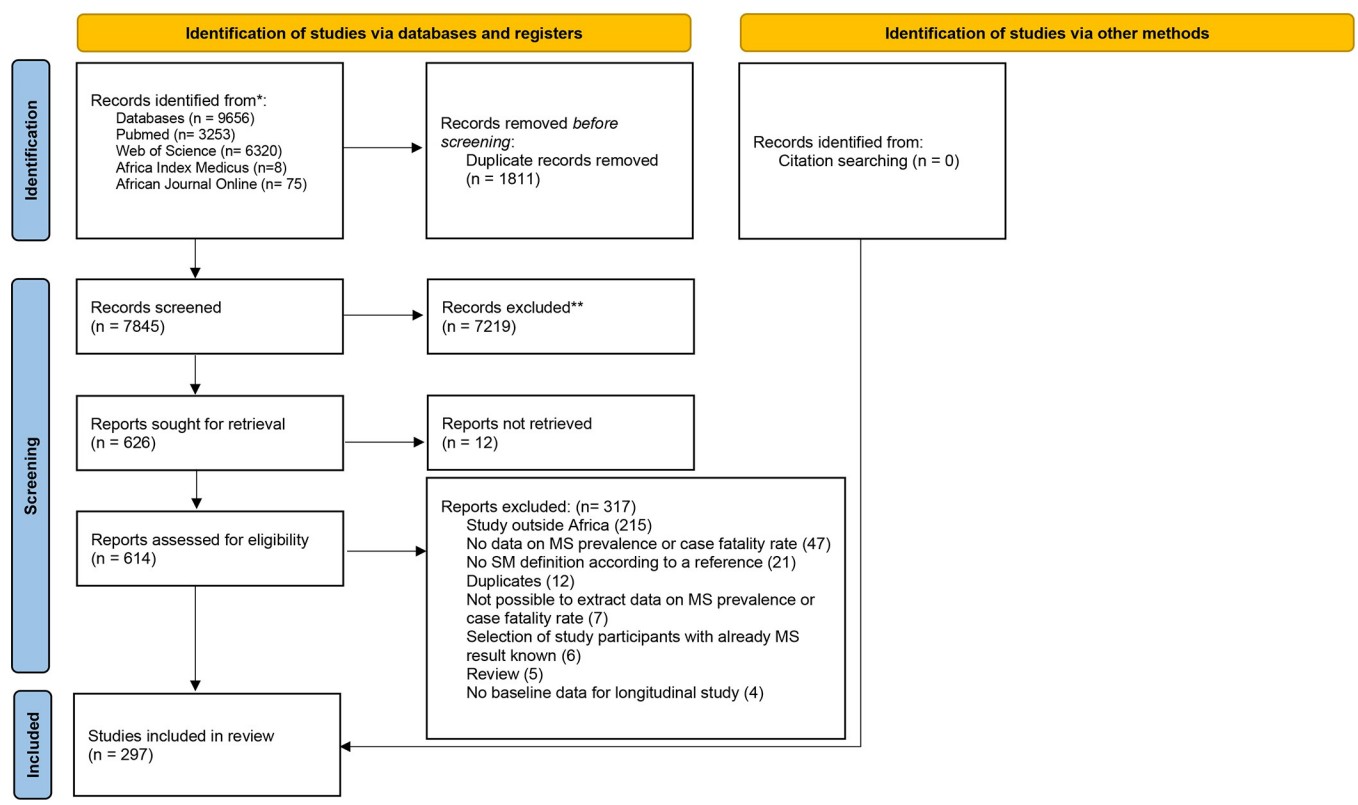

**Fig 1. PRISMA flow chart diagram showing study selection process.**

2001, 78 (22.6%) used the JIS 2009, 29 (8.4%) used the NCEP 2005 revised, 12 (3.5%) used the WHO 1998, and 2 (0.6%) used the AACCE 2003. Most of the prevalence data from the articles included in this review were free of bias (n = 174 prevalence data) (S6 Table).

## Prevalence of metabolic syndrome in Africa

MS prevalence was calculated using 345 prevalence data from 29 countries. In Africa, the overall pooled prevalence of MS was 32.4% [95% confidence interval (CI): 30.2–34.7] (Table 2).

## Prevalence of metabolic syndrome by diagnostic criteria

There was no significant difference in the pooled prevalence estimates between diagnostic criteria in this systematic review and meta-analysis (P = 0.168) (Figs 2 and S1). According to the

**Table 2. Summavry of meta-analysis results for the prevalence of metabolic syndrome in Africa.**

| | Prevalence. % (95%CI) | 95% Prediction interval | N Studies | N Participants | ¶H (95%CI) | §I² (95%CI) | P heterogeneity |
|---|---|---|---|---|---|---|---|
| **Metabolic syndrome** | | | | | | | |
| Overall | 32.4 [30.2–34.7] | [2.5–75.1] | 345 | 156464 | 9.5 [9.3–9.7] | 98.9 [98.9–98.9] | < 0.001 |
| Cross-sectional | 32.7 [30.4–35.1] | [2.5–75.5] | 318 | 152335 | 9.8 [9.6–10] | 99 [98.9–99] | < 0.001 |
| Low risk of bias | 32.2 [29.1–35.3] | [2.6–74.4] | 174 | 99136 | 10.5 [10.2–10.7] | 99.1 [99–99.1] | < 0.001 |

CI: Confidence interval; N: Number; 95% CI: 95% Confidence Interval; NA: Not applicable. ¶H is a measure of the extent of heterogeneity, a value of H = 1 indicates homogeneity of effects and a value of H >1indicates a potential heterogeneity of effects. §: I2 describes the proportion of total variation in study estimates that is due to heterogeneity, a value > 50% indicates presence of heterogeneity.

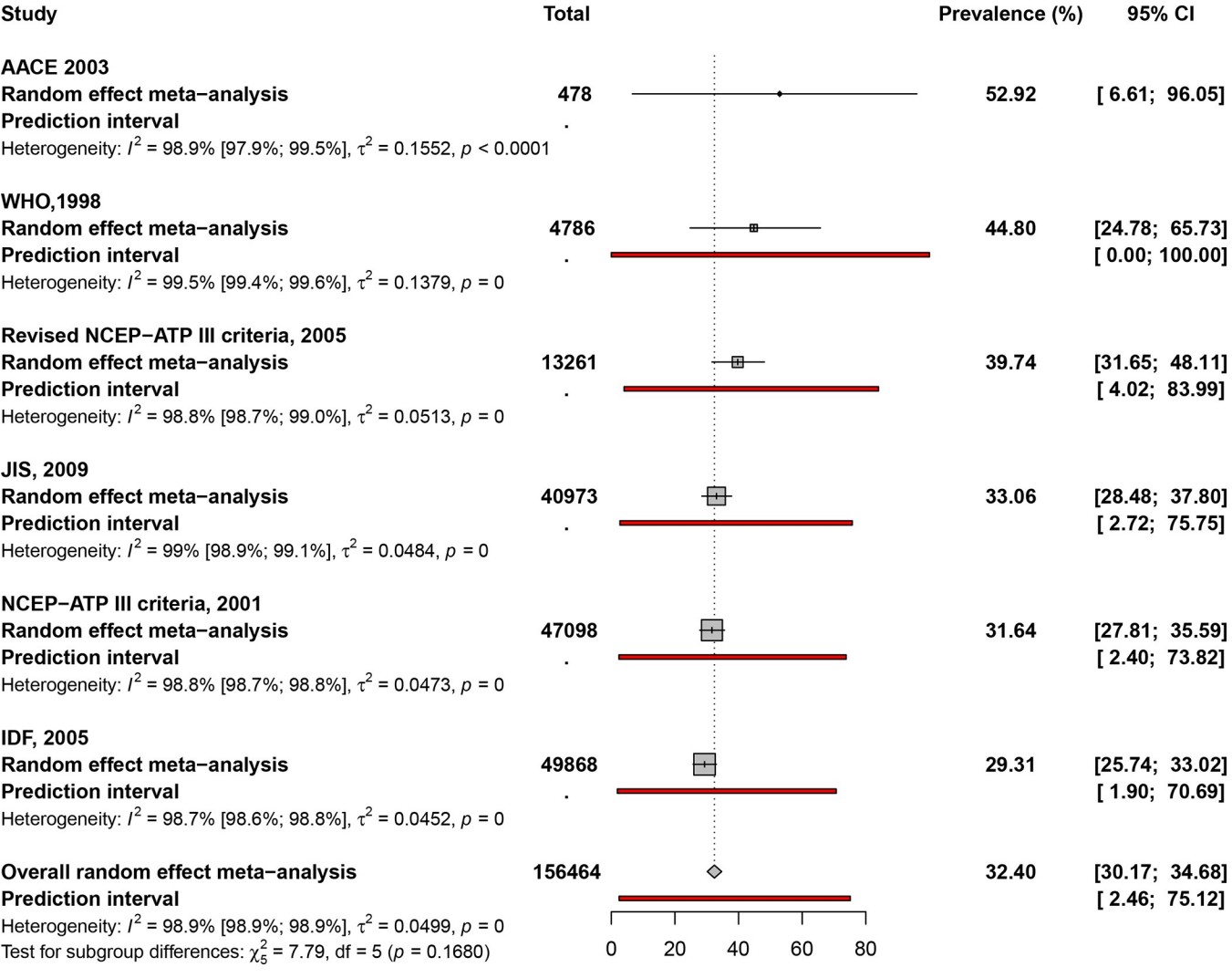

**Fig 2. Forest plot of showing prevalence of the metabolic syndrome in Africa.**

1998 WHO criterion, the pooled prevalence of MS was 44.8% (95% CI: 24.8–65.7). The combined prevalence of MS in studies that used the revised NCEP-ATP III 2005 criterion was 39.7% (95% CI: 31.7–48.1). The combined prevalence of MS was 33.1% (95% CI: 28.5–37.8) among studies that used the 2009 JIS criterion to diagnose MS. According to NCEP-ATP III 2001, the prevalence of diabetes in the African population was 31.6% (95% CI: 27.8–35.6). In the studies that used the IDF 2005, the combined prevalence of MS was 29.3% (95% CI: 25.7–33) (S7 Table). For all MS diagnostic criteria, there was significant heterogeneity in prevalence estimates across studies. ($I^2$ = 98.9%, p < 0.001).

## Prevalence of metabolic syndrome according to population categories

The combined prevalence of MS varied significantly by study population (p<0.001). Subgroup analysis (S7 Table) according to population groups shows that the reported prevalence tends to be higher for T2DM patients: 66.9% (95% CI: 60.3–73.1), followed by patients with coronary artery disease: 55.2% (95% CI: 50.8–59.6) and those with cardiovascular disease 48.3% (95% CI: 33.5–63.3). However, the lowest prevalence was obtained in psychiatric patients: 26.2%

(95% CI: 22.3–30.3), HIV-infected patients: 22.6% (95% CI: 18.6–26.9) and apparently healthy people: 22.2% (95% CI: 19.9–24.5). Heterogeneity in reported outcomes remained high in many subgroups (p < 0.001, except for patients with coronary artery disease (p = 0.956) and chronic musculoskeletal disease (p = 0.057).

### Prevalence of metabolic syndrome according to age

The overall prevalence of MS varied significantly by age group ((p<0.001)). Indeed, the overall pooled estimate for children aged 0–18 years was 13.3% (95% CI: 7.3–20.6), while for those aged 18 and up, it was 33.1% (95% CI: 30.8–35.5). There was significant variation among studies reporting prevalence in all age groups. (p<0.001) (S7 Table).

### Prevalence of metabolic syndrome according to gender

A total of 151 studies reported on the MS prevalence in the female population and 128 in the male population. The pooled prevalence of MS in women in Africa was 36.9% (95%CI: 33.2%-40.7%); while for men, the pooled prevalence was 26.7% (95%CI: 23.1%-30.5%), (p<0.001). There was significant heterogeneity between studies reporting prevalence in women and men (p<0.001).

### Prevalence of metabolic syndrome according to African Regions

Subgroup analysis revealed no statistically significant differences in MS burden by African region (p = 0.835) (Fig 3 and S7 Table). A total of 55 studies from the southern African region reported the burden of MS, with a combined prevalence of 33.6% (95% CI: 28.3–39.1). One hundred and twelve studies from West Africa looked at the prevalence of MS and found it to be 33.1% (95% CI: 28.7–37.7). A prevalence of 32.8% (95% CI: 28.6–37.3) was reported in 94 studies for North African countries. The pooled prevalence of MS in East and Central Africa was estimated to be 30.3% (95% CI: 25.6–35.2) and 30.1% (95% CI: 22.9–37.8) from 61 and 23 studies, respectively. Regardless of the African region of the USND, there was significant heterogeneity between studies reporting the prevalence of MS (p<0.001). The results of the subgroup analysis by country revealed a significant difference in prevalence (P<0.009), with Algeria having the highest prevalence, 43.9% (95% CI: 19.3–70.2), and Sudan having the lowest, 11.5% (95% CI: 5.1–19.9). We discovered no statistically significant differences in MS burden by WHO African region, country income level, or study design.

### Additional subgroup analyses

To obtain separate estimates, additional subgroup analysis was performed to investigate the source of heterogeneity. Thus, the prevalence of MS was significantly higher in studies conducted in a hospital setting (36.6%, 95%CI: 33.7–39.6, P<0.001), with a non-probabilistic sampling method (33.8%, 95%CI: 31.2–36.4, P = 0.007), and with samples collected prospectively (32.7%, 95%CI: 30.4–35, P = 0.045).

### Publication bias and sensitivity analysis

The funnel plots in S2 Fig show the publication bias between studies. The plots were asymmetric, indicating that there was significant publication bias. The Egger test was used to assess publication bias in all included studies. Overall, the analyses revealed a publication bias among the articles used to calculate MS prevalence in African countries (P Egger < 0.0001). The results of sensitivity analyses that included only cross-sectional studies and studies with a low risk of bias agreed well with the overall prevalence of MS (Table 2).

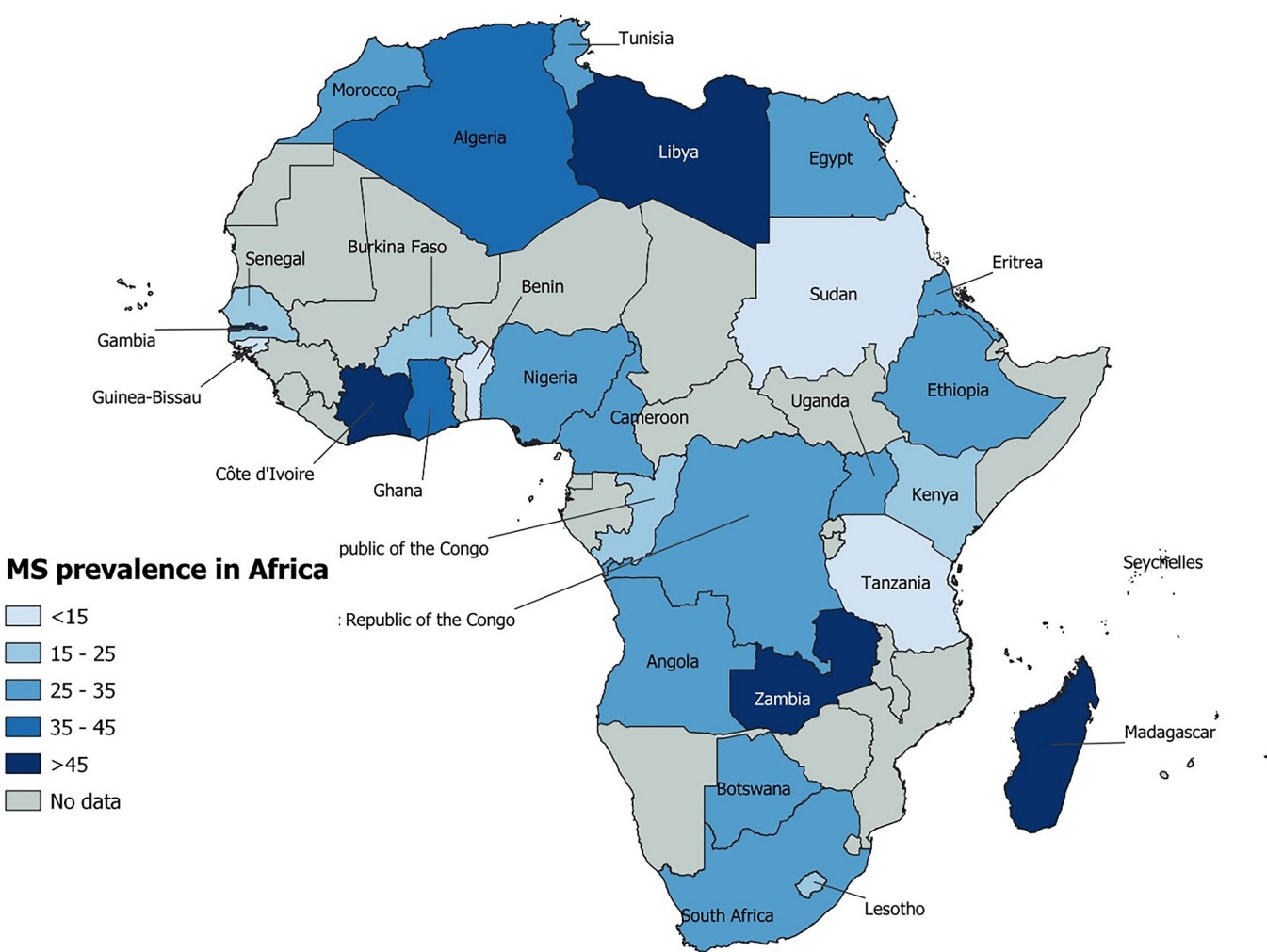

**Fig 3. Map of Africa indicating the prevalence of metabolic syndrome.**

## Discussion

We conducted this study to review available data and estimate the prevalence of MS in Africa. To the best of our knowledge, this is the first meta-analysis review to report the burden of MS in Africa and to present the pooled estimate of MS by definitional criteria and population groups. We examined data from 297 studies involving 156 464 participants from 29 African countries in total. The combined estimate of MS prevalence in the African population based on the data reviewed was 32.4%. We also recorded the age distribution, the criteria used to define MS, and the estimates by geographical region to see if there were any significant differences. Adults (over the age of 18) and patients with type 2 diabetes were the most vulnerable. According to subgroup analysis by country, the prevalence of MS was higher in Algeria (43.9%) than in other countries.

The prevalence of MS varies greatly depending on the diagnostic criteria used. The prevalence rates with WHO were 44.8%, 39.7% with revised NCEP-ATP, 33.1% with JIS, and 31.6% with JIS. NCEP-ATP scored 29.3%, while IDF scored 29.3%. Although no statistically significant difference was found, the higher prevalence obtained with the WHO definition in our meta-analysis could be attributed to the fact that, unlike the other definitions, this definition

includes BMI as a diagnostic criterion for MS, which is the most important risk factor in the African population [339]. Overall, our findings differ from those of many other studies, including one conducted in the global population, where the prevalence of diabetes was 16.0%, 20.9%, 23.9%, 25.4%, 28.2%, 29.1%, and 31.4%, respectively, according to EGIR, WHO, AACE, ATP III-, IDF, revised ATP-III, and JIS criteria [340]. Another study in South Asia discovered a 14.0% (WHO), 26.1% (ATP III), 29.8% (IDF), and 32.5% (modified ATP III) prevalence of MS [341]. This disparity between the results of different studies from around the world is not surprising, as recent reviews have made the same finding and unanimously suggest the importance of adopting a standardised approach to MS classification to allow better comparability between countries in general [341–343].

According to the data compiled in this review, the prevalence of MS was higher (66.9%) in T2DM patients. Given that T2DM is an entity of MS, this high prevalence rate is not surprising [344]. This observation may be supported by the fact that the WHO definition that reported the highest prevalence in our review includes diabetes as a criterion for the diagnosis of MS. The findings of this review agree with those of Shiferaw et al [29] and confirm that the high prevalence rates of MS in the African population are more prevalent in T2DM patients. This emphasizes the significance of strengthening MS control strategies in these patients to protect them from developing cardiovascular disease, which is the most common complication of MS.

MS was more common in people over the age of 18. It has already been demonstrated that the components of MS can be found in adults and are associated with aging [345]. However, a significant rate of MS was observed in the under-18 age group (13.3%), which is consistent with previous studies that reported that the prevalence of MS in this age group ranged from 1.2% to 22.6% [346] and from 0% to 19.2% [347], irrespective of the diagnostic methods. The high prevalence of MS in the under-18 age group may be explained by the increasing early development of obesity, which is the primary metabolic crossroads that predisposes to the development of MS. Indeed, according to a 2017 WHO study, the prevalence of overweight and obesity among children and adolescents in Africa was 16.5% in 2017, or 9.7 million cases [348]. This high prevalence of childhood obesity has most likely contributed to the African continent's high prevalence of MS. In general, special consideration should be given to all age groups of the population in the management of MS component diagnosis and treatment.

The gender distribution of MS prevalence showed that women had a higher prevalence (36.9%) than men (26.9%). Previous studies conducted on the African continent (Jeaspers et al., 2020; ofori-asenso et al., 2017; Ambachew et al., 2020) have shown similar results [349–351]. This could be explained by the obesity epidemic, which is a major risk factor for MS and predominantly affects women [352]. Menopause, polycystic ovarian syndrome, and low physical activity are additional risk factors that could explain this high prevalence in this population group.

This study's data show no statistically significant difference in the prevalence of MS between sub-regions. This demonstrates that the MS epidemic does not spare any African subregion, and thus all African regions would face the same major health challenges.

The study's main strength is that it provides data on the overall epidemiology of MS in the African population. Furthermore, one of the process's strengths is its comprehensiveness, which included a search of four different databases, well-defined inclusion/exclusion criteria, and extensive use of reference lists. Although generalization should be exercised with caution, we believe that the pooled MS prevalence is representative of Africa, as populations from all five African regions (central, eastern, southern, northern, and western) were included. The pooled MS prevalence presented here should be a useful starting point for understanding the disease burden in Africa and guiding future lines of research, prevention design, and resource allocation and planning.

There are some limitations to this study that should be considered in future research. First, the various types of definitions used to diagnose MS in the included studies may have an impact on the pooled prevalence estimate, which may differ from the true prevalence in Africa obtained in this study. Furthermore, the MS diagnostic criteria examined were not specific to Africa. Indeed, the lack of ethnic cut-offs for waist circumference specific to the African population forces studies conducted on the continent to rely on available definitions based on Western population cut-offs. Despite our extensive efforts to identify and access MS data in all African countries, some countries had no or few data sources, potentially contributing to an underestimation of the true burden of SM in Africa.

## Conclusion

According to the findings of this study, the prevalence of MS is high in Africa. Type 2 diabetic patients and populations over the age of 18 are the groups most at risk in Africa, indicating the need for immediate clinical and public health attention. More information on the clinical criteria of the MS components and their interrelationships will allow for a better understanding of the underlying factors causing this disease in the African population, allowing for the implementation of appropriate protective measures, such as lifestyle changes, to improve them.

## Supporting information

**S1 Fig. Forest plot of metabolic syndrome prevalence in Africa based on diagnostic criteria.**
(PDF)

**S2 Fig. Funnel chart for publications of the prevalence of metabolic syndrome in Africa.**
(PDF)

**S1 Table. Preferred reporting items for systematic reviews and meta-analyses checklist.**
(PDF)

**S2 Table. Search strategy used for searching articles from PubMed.**
(PDF)

**S3 Table. Items for risk of bias assessment.**
(PDF)

**S4 Table. Characteristics of included studies that reported of prevalence of metabolic syndrome in Africa.**
(PDF)

**S5 Table. Individual characteristics of included studies.**
(PDF)

**S6 Table. Risk of bias assessment.**
(PDF)

**S7 Table. Subgroup analyses of prevalence of metabolic syndrome in African studies.**
(PDF)

## Author Contributions

**Conceptualization:** Arnol Bowo-Ngandji, Sebastien Kenmoe, Judith Laure Ngondi.

**Data curation:** Arnol Bowo-Ngandji, Sebastien Kenmoe, Jean Thierry Ebogo-Belobo, Raoul Kenfack-Momo, Guy Roussel Takuissu, Cyprien Kengne-Ndé, Donatien Serge Mbaga, Serges Tchatchouang, Josiane Kenfack-Zanguim, Robertine Lontuo Fogang, Elisabeth Zeuko'o Menkem, Juliette Laure Ndzie Ondigui, Ginette Irma Kame-Ngasse, Jeannette Nina Magoudjou-Pekam, Maxwell Wandji Nguedjo.

**Formal analysis:** Sebastien Kenmoe, Cyprien Kengne-Ndé.

**Funding acquisition:** Sebastien Kenmoe.

**Methodology:** Arnol Bowo-Ngandji, Sebastien Kenmoe, Jean Thierry Ebogo-Belobo, Raoul Kenfack-Momo, Guy Roussel Takuissu, Cyprien Kengne-Ndé, Donatien Serge Mbaga, Serges Tchatchouang, Josiane Kenfack-Zanguim, Robertine Lontuo Fogang, Elisabeth Zeuko'o Menkem, Juliette Laure Ndzie Ondigui, Ginette Irma Kame-Ngasse, Jeannette Nina Magoudjou-Pekam, Maxwell Wandji Nguedjo, Jean Paul Assam Assam, Damaris Enyegue Mandob, Judith Laure Ngondi.

**Project administration:** Sebastien Kenmoe, Judith Laure Ngondi.

**Supervision:** Sebastien Kenmoe.

**Validation:** Arnol Bowo-Ngandji, Sebastien Kenmoe, Jean Thierry Ebogo-Belobo, Raoul Kenfack-Momo, Guy Roussel Takuissu, Cyprien Kengne-Ndé, Donatien Serge Mbaga, Serges Tchatchouang, Josiane Kenfack-Zanguim, Robertine Lontuo Fogang, Elisabeth Zeuko'o Menkem, Juliette Laure Ndzie Ondigui, Ginette Irma Kame-Ngasse, Jeannette Nina Magoudjou-Pekam, Maxwell Wandji Nguedjo, Jean Paul Assam Assam, Damaris Enyegue Mandob, Judith Laure Ngondi.

**Writing – original draft:** Arnol Bowo-Ngandji, Sebastien Kenmoe.

**Writing – review & editing:** Arnol Bowo-Ngandji, Sebastien Kenmoe, Jean Thierry Ebogo-Belobo, Raoul Kenfack-Momo, Guy Roussel Takuissu, Cyprien Kengne-Ndé, Donatien Serge Mbaga, Serges Tchatchouang, Josiane Kenfack-Zanguim, Robertine Lontuo Fogang, Elisabeth Zeuko'o Menkem, Juliette Laure Ndzie Ondigui, Ginette Irma Kame-Ngasse, Jeannette Nina Magoudjou-Pekam, Maxwell Wandji Nguedjo, Jean Paul Assam Assam, Damaris Enyegue Mandob, Judith Laure Ngondi.

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
