## [Decision Letter · Decision Letter 0]

14 Mar 2023

PONE-D-22-31497Prevalence of metabolic syndrome in African population: a systematic review and meta-analysisPLOS ONE

Dear Dr. Kenmoe,

Thank you for submitting your manuscript to PLOS ONE. After careful consideration, we feel that it has merit but does not fully meet PLOS ONE’s publication criteria as it currently stands. Therefore, we invite you to submit a revised version of the manuscript that addresses the points raised during the review process.

Dear Authors,-Please address to the quality assessment checklist in the abstract.-Please replace PRISMA 2009 with PRISMA flowchart 2020, in the main manuscript, figure 1, and ref. list:

Page, M.J., McKenzie, J.E., Bossuyt, P.M., Boutron, I., Hoffmann, T.C., Mulrow, C.D., et al., 2021. The PRISMA 2020 statement: an updated guideline for reporting systematic reviews. BMJ. 372:n71. doi: 10.1136/bmj.n71<o:p></o:p>

-It is better to summarize reasons of excluded full texts-In PRISMA checklist (Table S1)  should be inserted just page number, not location or other data.

We look forward to receiving your revised manuscript.

Kind regards,

Ozra Tabatabaei-Malazy

Academic Editor

PLOS ONE

Journal Requirements:

2. We note that Figure 3 in your submission contain [map/satellite] images which may be copyrighted. All PLOS content is published under the Creative Commons Attribution License (CC BY 4.0), which means that the manuscript, images, and Supporting Information files will be freely available online, and any third party is permitted to access, download, copy, distribute, and use these materials in any way, even commercially, with proper attribution. For these reasons, we cannot publish previously copyrighted maps or satellite images created using proprietary data, such as Google software (Google Maps, Street View, and Earth). For more information, see our copyright guidelines: http://journals.plos.org/plosone/s/licenses-and-copyright.

a. You may seek permission from the original copyright holder of Figure 3 to publish the content specifically under the CC BY 4.0 license.  

3. We note that this manuscript is a systematic review or meta-analysis; our author guidelines therefore require that you use PRISMA guidance to help improve reporting quality of this type of study. Please upload copies of the completed PRISMA checklist as Supporting Information with a file name “PRISMA checklist”.

Reviewers' comments:

Reviewer's Responses to Questions

**Comments to the Author**

1. Is the manuscript technically sound, and do the data support the conclusions?

Reviewer #1: Yes

Reviewer #2: Yes

2. Has the statistical analysis been performed appropriately and rigorously? 

Reviewer #1: Yes

Reviewer #2: Yes

3. Have the authors made all data underlying the findings in their manuscript fully available?

Reviewer #1: Yes

Reviewer #2: Yes

4. Is the manuscript presented in an intelligible fashion and written in standard English?

Reviewer #1: Yes

Reviewer #2: Yes

5. Review Comments to the Author

Reviewer #1: In this manuscript, Bowo-Ngandji et al. conducted a systematic review and meta-analysis to assess the prevalence of metabolic syndrome in African population. They separately analyzed studies using the WHO, revised NCEP-ATP III, JIS, NCEP/ATP III and IDF definition criteria. This is a high-quality and comprehensive study. The statistical analysis sounds appropriate.

I only have a few minor comments:

1- In the abstract, the authors mention “and type 2 diabetics were the most affected,with a prevalence of 66.9%”. The comparison seems to be incomplete.

2- I suggest that authors spell out numbers smaller than ten in the introduction.

3- It should be “Joint Interim Statement not” “Join Interim Statement”.

Reviewer #2: Examining the overall prevalence of metabolic syndrome in Africa has been previously reported in various studies as the authors have mentioned, but overall reporting and meta-analysis have not been done before, which is important in this study.

It would be better to report the prevalence by gender, too.

In the distribution of countries, it is good to include the names of the countries.

6. PLOS authors have the option to publish the peer review history of their article (what does this mean?). If published, this will include your full peer review and any attached files.

Reviewer #1: No

Reviewer #2: No

---

## [Author Response · Author response to Decision Letter 0]

1 Jun 2023

Review Comments to the Author

Reviewer #1: In this manuscript, Bowo-Ngandji et al. conducted a systematic review and meta-analysis to assess the prevalence of metabolic syndrome in African population. They separately analyzed studies using the WHO, revised NCEP-ATP III, JIS, NCEP/ATP III and IDF definition criteria. This is a high-quality and comprehensive study. The statistical analysis sounds appropriate. 

Authors: We thank the reviewer for this appreciation.

1. In the abstract, the authors mention “and type 2 diabetics were the most affected,with a prevalence of 66.9%”.The comparison seems to be incomplete. 

Authors: Thank you for your valuable feedback. We agree with your observation and have now completed the comparison in abstract. 

2. I suggest that authors spell out numbers smaller than ten in the introduction.

Authors: Thank you for your valuable feedback. We agree with your suggestion and have now included in the introduction.

3. It should be “Joint Interim Statement not” “Join Interim Statement”

Authors: Thank you for your valuable feedback. We agree with your suggestion and have now included on line 79

Reviewer #2: Examining the overall prevalence of metabolic syndrome in Africa has been previously reported in various studies as the authors have mentioned, but overall reporting and meta-analysis have not been done before, which is important in this study. 

Authors: We thank the reviewer for this summary and appreciation.

1. It would be better to report the prevalence by gender, too. Authors: Thank you for this valuable feedback on our manuscript. We have provided more information regarding gender distribution in abstract, results, and discussion.

2. In the distribution of countries, it is good to include the names of the countries. 

Authors: Thank you for your thorough review and for bringing to our attention to include the names of the countries in the distribution of countries in our article. We have revised our manuscript to include all the names of the countries in the distribution of countries.

---

## [Decision Letter · Decision Letter 1]

28 Jun 2023

PONE-D-22-31497R1Prevalence of metabolic syndrome in African population: a systematic review and meta-analysisPLOS ONE

Dear Dr. Kenmoe,

Thank you for submitting your manuscript to PLOS ONE. After careful consideration, we feel that it has merit but does not fully meet PLOS ONE’s publication criteria as it currently stands. Therefore, we invite you to submit a revised version of the manuscript that addresses the points raised during the review process.

We look forward to receiving your revised manuscript.

Kind regards,

Ozra Tabatabaei-Malazy

Academic Editor

PLOS ONE

Journal Requirements:

Additional Editor Comments:

Section Editor# comments:

Dear Authors,

-Please address to the quality assessment checklist in the abstract.

-Please replace PRISMA 2009 with PRISMA flowchart 2020, in the main manuscript, figure 1, and ref. list:

Page, M.J., McKenzie, J.E., Bossuyt, P.M., Boutron, I., Hoffmann, T.C., Mulrow, C.D., et al., 2021. The PRISMA 2020 statement: an updated guideline for reporting systematic reviews. BMJ. 372:n71. doi: 10.1136/bmj.n71

-It is better to summarize reasons of excluded full texts

-In PRISMA checklist (Table S1) should be inserted just page number, not location or other data.

Best Regards,

Reviewers' comments:

Reviewer's Responses to Questions

**Comments to the Author**

1. If the authors have adequately addressed your comments raised in a previous round of review and you feel that this manuscript is now acceptable for publication, you may indicate that here to bypass the “Comments to the Author” section, enter your conflict of interest statement in the “Confidential to Editor” section, and submit your "Accept" recommendation.

Reviewer #1: All comments have been addressed

Reviewer #2: All comments have been addressed

2. Is the manuscript technically sound, and do the data support the conclusions?

Reviewer #1: Yes

Reviewer #2: Yes

3. Has the statistical analysis been performed appropriately and rigorously? 

Reviewer #1: Yes

Reviewer #2: Yes

4. Have the authors made all data underlying the findings in their manuscript fully available?

Reviewer #1: Yes

Reviewer #2: Yes

5. Is the manuscript presented in an intelligible fashion and written in standard English?

Reviewer #1: Yes

Reviewer #2: Yes

6. Review Comments to the Author

Reviewer #1: I thank the authors for their detailed response. All my comments have been addressed and I no further comments.

Reviewer #2: The authors have adequately addressed my comments and the submitted manuscript is acceptable for publication.

7. PLOS authors have the option to publish the peer review history of their article (what does this mean?). If published, this will include your full peer review and any attached files.

Reviewer #1: No

Reviewer #2: No

---

## [Author Response · Author response to Decision Letter 1]

6 Jul 2023

July, 04th 2023

Sebastien KENMOE, PhD

Department of Microbiology and Parasitology

University of Buea

e mail : sebastien.kenmoe@ubuea.cm

Tél: + (237) 674 05 95 26

Submission Manuscript ID PONE-D-22-31497

Article title: Prevalence of the metabolic syndrome in African populations: a systematic review and meta-analysis.

The Editor-in-Chief

Plos One

Dear Editor, 

We are delighted that the Plos One will consider publication of our paper, pending satisfactory revisions as suggested by the reviewers. 

We have given careful consideration to the reviewer’s comments and have done our best to address them all. 

In blue text in the response to the reviewer comment document is a point-by-point explanation of how we have addressed the concerns and revised our manuscript. We have enclosed 

1. Point-by-point response letter

2. Track-changes version of our revised manuscript

3. A clean version of our revised manuscript

We thank the reviewers for their thoughtful comments. With these revisions, we feel the paper has been substantially improved. We hope it will receive favorable consideration for publication in the Plos One. 

Please do not hesitate to contact us should you have any additional questions or comments. 

Kind regards,

Sebastien Kenmoe 

On behalf of the co-authors

Review Comments to the Author

Editor #:

1. Please address to the quality assessment checklist in the abstract.

Authors: Thank you for this valuable feedback on our manuscript. We have now added the method for assessing risk of bias in the abstract.

2. Please replace PRISMA 2009 with PRISMA flowchart 2020, in the main manuscript, figure 1, and ref. list.

Authors: Thank you for this valuable feedback on our manuscript. Your comment has been incorporated, and the flowchart and figure 1 have been updated with the PRISMA 2020 reference.

3. It is better to summarize reasons of excluded full texts

Authors: Thank you for your valuable feedback. We agree with your suggestion and have now included in Figure 1.

4. In PRISMA checklist (Table S1) should be inserted just page number, not location or other data.

Authors: Thank you for your valuable feedback. Your suggestion had already been taken into account in S1 Table.

---

## [Editor Report · Decision Letter 2]

9 Jul 2023

PONE-D-22-31497R2Prevalence of metabolic syndrome in African population: a systematic review and meta-analysisPLOS ONE

Dear Dr. Kenmoe,

Thank you for submitting your manuscript to PLOS ONE. After careful consideration, we feel that it has merit but does not fully meet PLOS ONE’s publication criteria as it currently stands. Therefore, we invite you to submit a revised version of the manuscript that addresses the points raised during the review process.

ACADEMIC EDITOR:

Dear Authors,Please consider following comments:1-Replace the Ref. 32 with following Ref.:Page, M.J., McKenzie, J.E., Bossuyt, P.M., Boutron, I., Hoffmann, T.C., Mulrow, C.D., et al., 2021. The PRISMA 2020 statement: an updated guideline for reporting systematic reviews. BMJ. 372:n71. doi: 10.1136/bmj.n712- Please re-calculate records in PRISMA flowchart. Best Regards,

We look forward to receiving your revised manuscript.

Kind regards,

Ozra Tabatabaei-Malazy

Academic Editor

PLOS ONE
---

## [Author Response · Author response to Decision Letter 2]

10 Jul 2023

Dear Editor,

Thank you for your comments.

1. We have updated reference 32. 

2. We have re-calculated the records in the PRISMA flowchart (Fig 1). 

We sincerely thank you for your time and attention to detail.

Best regards,

Sebastien Kenmoe

---

## [Editor Report · Decision Letter 3]

11 Jul 2023

PONE-D-22-31497R3Prevalence of metabolic syndrome in African population: a systematic review and meta-analysisPLOS ONE

Dear Dr. Kenmoe,

Thank you for submitting your manuscript to PLOS ONE. After careful consideration, we feel that it has merit but does not fully meet PLOS ONE’s publication criteria as it currently stands. Therefore, we invite you to submit a revised version of the manuscript that addresses the points raised during the review process.

ACADEMIC EDITOR:-Please clarify where are 12 records in screening section of the PRISMA flowchart? (difference between 626 sought for reterival and 614 assessed for eligibility)-Correct the Ref.32 accordance with method of address writing in "cite" section of the PubMed.

We look forward to receiving your revised manuscript.

Kind regards,

Ozra Tabatabaei-Malazy

Academic Editor

PLOS ONE
---

## [Author Response · Author response to Decision Letter 3]

11 Jul 2023

Dear Editor,

Thank you for pointing out the discrepancies in the PRISMA flowchart and the issue with Ref.32.

We corrected the issues.

Regards,

Sebastien Kenmoe

---

## [Editor Report · Decision Letter 4]

13 Jul 2023

Prevalence of metabolic syndrome in African population: a systematic review and meta-analysis

PONE-D-22-31497R4

Dear Dr. Kenmoe,

We’re pleased to inform you that your manuscript has been judged scientifically suitable for publication and will be formally accepted for publication once it meets all outstanding technical requirements.

Kind regards,

Ozra Tabatabaei-Malazy

Academic Editor

PLOS ONE
---

## [Editor Report · Acceptance letter]

17 Jul 2023

PONE-D-22-31497R4 

Prevalence of the metabolic syndrome in African populations: a systematic review and meta-analysis

Dear Dr. Kenmoe:

I'm pleased to inform you that your manuscript has been deemed suitable for publication in PLOS ONE. Congratulations! Your manuscript is now with our production department. 

Kind regards, 

on behalf of

Dr. Ozra Tabatabaei-Malazy 

Academic Editor

PLOS ONE